

# An efficient hybrid differential evolution-golden jackal optimization algorithm for multilevel thresholding image segmentation

Xianmeng Meng[1,2], Linglong Tan[1] and Yueqin Wang[1]

[1] School of Electronics Engineering, Anhui Xinhua University, Hefei, China
[2] School of Computer Science and Information Engineering, Hefei University of Technology, Hefei, China

## ABSTRACT

Image segmentation is a crucial process in the field of image processing. Multilevel threshold segmentation is an effective image segmentation method, where an image is segmented into different regions based on multilevel thresholds for information analysis. However, the complexity of multilevel thresholding increases dramatically as the number of thresholds increases. To address this challenge, this article proposes a novel hybrid algorithm, termed differential evolution-golden jackal optimizer (DEGJO), for multilevel thresholding image segmentation using the minimum cross-entropy (MCE) as a fitness function. The DE algorithm is combined with the GJO algorithm for iterative updating of position, which enhances the search capacity of the GJO algorithm. The performance of the DEGJO algorithm is assessed on the CEC2021 benchmark function and compared with state-of-the-art optimization algorithms. Additionally, the efficacy of the proposed algorithm is evaluated by performing multilevel segmentation experiments on benchmark images. The experimental results demonstrate that the DEGJO algorithm achieves superior performance in terms of fitness values compared to other metaheuristic algorithms. Moreover, it also yields good results in quantitative performance metrics such as peak signal-to-noise ratio (PSNR), structural similarity index (SSIM), and feature similarity index (FSIM) measurements.

## INTRODUCTION

Image segmentation is a crucial step in image processing, and the correctness of image segmentation directly affects the extraction, detection, and recognition of objects (*Pare et al., 2019*). According to the features of the image, such as grayscale, histogram, and geometric shape, the image is segmented into independent regions to obtain reduce the complexity of image analysis. It has been widely used in remote sensing, medicine, and surveillance fields (*Elaziz, Ewees & Oliva, 2020*; *Houssein et al., 2021a*).

Recently, multiple methods for image segmentation have been presented, which can be categorized into (1) histogram thresholding-based methods, (2) region feature-based methods, (3) clustering-based techniques, (4) texture feature-based methods, and (5)

Corresponding author
Linglong Tan,
tanlinglong@axhu.edu.cn

artificial intelligence techniques (*Ayala et al., 2023*; *Bhargavi & Jyothi, 2014 Rai, Das & Dhal, 2022*; *Zhang et al., 2021*). Histogram thresholding-based methods utilize histogram data to obtain segmentation thresholds and divide the image pixels into multiple independent regions. The methods are widely applied to image segmentation techniques due to their simplicity, accuracy, and robustness (*Elaziz, Ewees & Oliva, 2020*).

Approaches based on image histogram thresholding can be classified as bi-level and multilevel (*Pare et al., 2019*; *Wunnava et al., 2022*). Bi-level thresholding methods distinguish the region of interest from the image background using a single threshold. On the contrary, in the multilevel thresholding methods, the images are divided into several regions with multiple threshold values. For the processing of real images, researchers may require information from multiple regions, and bi-level thresholding does not meet the requirements. Therefore, the histogram of an image needs to be divided into regions by multilevel threshold segmentation. In multilevel threshold segmentation, the main methods for determining the optimal threshold are: parametric and non-parametric (*Wang et al., 2023a*). Parametric approaches define each category of the image by determining the probability density distribution of the image region. However, these methods are complex to implement and the estimation of parameters is limited by the initial conditions. On the other hand, non-parametric approaches employ discriminative rules to partition image pixels into homogeneous regions and then determine thresholds based on criteria of entropy or variance (*Wang et al., 2023a*). Over the years, various criteria have been developed, including the Otsu (*Otsu, 1975*), Kapur entropy (*Kapur, Sahoo & Wong, 1985*), and minimum cross entropy (MCE) (*Li & Lee, 1993*) criteria, aimed at identifying optimal thresholds for segmented images. Nevertheless, these approaches have some limitations, such as the high computational complexity of these methods, especially for high threshold numbers.

For image segmentation, the determination of multilevel thresholding could be considered as an NP-hard optimization problem, and the objective functions are chosen as the Otsu, Kapur entropy and MCE (*Rodríguez-Esparza et al., 2020*). The optimal multilevel threshold is typically derived through iterative optimization algorithms.

Metaheuristic algorithms (MAs) are widely used for solving NP-hard optimization problems (*Houssein et al., 2021a*). MA methods are typically inspired by natural processes, which could be divided into three categories: (1) evolution-based algorithms, (2) physics-inspired algorithms, and (3) swarm intelligence algorithms. More specifically, evolution-based algorithms are inspired by the laws of natural evolution, including genetic algorithms (GA) (*Kazarlis, Bakirtzis & Petridis, 1996*), and differential evolutionary (DE) Algorithms (*Storn & Price, 1997*). Physically-inspired algorithms are based on modeling various observed physical phenomena. Such algorithms include the spectral optimizer (LSO) (*Abdel-Basset et al., 2022*), water flow optimizer (WFO) (*Luo, 2022*), geometric mean optimizer (GMO) (*Rezaei et al., 2023*), and many others. Swarm intelligence algorithms are based on modeling group behavior in nature, such as grey wolf optimizer (GWO) (*Mirjalili, Mirjalili & Lewis, 2014*), wild horse optimizer (WHO) (*Naruei & Keynia, 2021*), whale optimization algorithm (WOA) (*Mirjalili & Lewis, 2016*), Harris hawks optimization

(HHO) (*Heidari et al., 2019*), horned lizard optimization (HLO) algorithm (*Peraza-Vázquez et al., 2024*), and so on.

To achieve an optimal multilevel threshold, many MA algorithms have been presented to address the multilevel thresholding problem (*Sharma et al., 2020*; *Tan & Zhang, 2020*). Specifically, *Ahmadi et al. (2019)* presented a bird mating optimization (BMO) for image segmentation using Otsu and Kapur techniques. The performance of the BMO approach was demonstrated compared to existing algorithms in terms of segmentation quality. *Rodríguez-Esparza et al. (2020)* developed an efficient HHO algorithm for multilevel image segmentation utilizing the MCE as the objective function. The results indicated the HHO algorithm's superior performance with benchmark images. *Eisham et al. (2022)* introduced a chimp optimization algorithm (COA) for image segmentation, outperforming existing methods in segmenting benchmark images. *Wang & Song (2022)* proposed an image segmentation method with an adaptive firefly algorithm (AFA), using the MCE method as an objective function. The AFA algorithm exhibited excellent segmentation quality and low computation time. *Houssein et al. (2021a)*; *Houssein et al. (2021b)* proposed a black widow optimization (BWO) algorithm to determine optimal thresholds for multilevel thresholding. Furthermore, *Houssein et al. (2023)* enhanced the heap-based optimizer (EHBO) and applied it to multi-threshold segmentation. The results indicated that the EHBO algorithm has superior image segmentation performance.

As mentioned above, the implementation of meta-heuristic algorithms for thresholding images could reduce the complexity of finding the optimal threshold. Not all meta-heuristic optimization algorithms could solve this problem according to the 'No Free Lunch (NFL)' theorem proposed by *Wolpert & Macready (1997)*. Consequently, it is necessary to perform tests on multi-threshold segmentation methods for images.

As a novel metaheuristic algorithm, Golden Jackal Optimization (GJO) was presented based on the social and hunting behaviors of golden jackals (*Chopra & Mohsin Ansari, 2022*), which has the advantages of simplicity of principles, fewer parameter requirements, and excellent performance. Compared to other metaheuristic algorithms, the GJO has been demonstrated to be an effective optimization method. Similar to other MA algorithms, the GJO algorithm has some limitations, the main limitation is that GJO tends to fall into local optima with solving practical problems (*Houssein et al., 2022*; *Zhang et al., 2023*).

As one of the earliest metaheuristic algorithms, the DE algorithm (*Storn & Price, 1997*) has been widely used in several fields due to its simple structure. However, the DE algorithm suffers from poor performance in solving complex optimization problems, which has led many researchers to improve it to obtain superior performance. Because the DE algorithm is easily combined with other algorithms, the main improvement of the DE algorithm is to combine it with other meta-heuristic algorithms to enhance the optimization performance.

To overcome the problem of low convergence accuracy, a novel hybrid algorithm based on differential evolution (DE) and GJO algorithms (DEGJO) is proposed, which utilizes the DE algorithm to update the optimal locations of the jackal to enhance the search performance. The DEGJO is used for image segmentation based on the MCE method, which aims to achieve threshold and minimum fitness values. The stability and adaptability of the DEGJO are demonstrated through benchmark function experiments,

and the segmentation results are compared with those of original DE, GWO, WOA, GJO, and hybrid GJO (HGJO) algorithms for minimum cross entropy. To evaluate the quality of image segmentation, several evaluation metrics are employed. The experimental results of benchmark functions and images show that the DEGJO obtain good effectiveness and stability.

The main contributions of this article are as follows.

(1) A novel hybrid DEGJO is presented by improving the optimal position update in GJO through the differential position and selection operators of the DE algorithm.

(2) The performance of the DEGJO is evaluated using the CEC2021 benchmark function, and the DEGJO is compared with other MA algorithms: original DE, GWO, WOA, GJO, and hybrid GJO.

(3) The DEGJO is applied to solve the image segmentation problem using the minimum cross-entropy function. The performance of the proposed algorithm is validated on different multilevel segmentation experiments, and the DEGJO algorithm achieves superior performance compared to different MA algorithms.

This article is organized as follows. 'Material and Methods' introduces the MCE, the GJO algorithm, the DE algorithm, and the DEGJO algorithm. Test function and image segmentation experiments are performed in 'Experimental Analysis and Discussion'. Finally, the conclusions and future work are presented in 'Conclusions'.

# MATERIALS & METHODS

## Minimum cross-entropy in multilevel image thresholding

It is effective and convenient to obtain image information through image segmentation. Based on the features of the image, the image is segmented into multiple regions, thus presenting distinctive features of different regions. The histogram provides a visual representation of the distribution of pixels throughout the image. Consequently, image histogram segmentation is achieved by determining the threshold value using the MCE method (*Wang & Song, 2022*).

The cross-entropy can be characterized using the information-theoretic distance between the two probability distributions $P = [p_1, p_2, \ldots, p_N]$ and $Q = [q_1, q_2, \ldots, q_N]$, expressed as (*Rodríguez-Esparza et al., 2020*):

$$D(P, Q) = \sum_{N}^{i=1} p_i \log \frac{p_i}{q_i}, \tag{1}$$

Using the cross-entropy as the objective function, the optimal thresholding is determined by minimizing the cross-entropy.

According to the image histogram $I$, the segmented image can be computed as:

$$I_{seg}(x, y) = \begin{cases} v(1, th) = \sum_{i=1}^{th-1} ih(i) \bigg/ \sum_{i=1}^{th-1} h(i), & \text{if } I(x, y) < th \\ v(th, L+1) = \sum_{i=th}^{L} ih(i) \bigg/ \sum_{i=th}^{L} h(i), & \text{if } I(x, y) \geq th \end{cases}, \tag{2}$$

where $h(i)$ is the histogram, $L$ represents the gray level, and $th$ is the threshold number.

The formula for the MCE is calculated as:

$$f_{cross}(th) = \sum_{L}^{i=1} ih(i)\log(i) - \sum_{th-1}^{i=1} ih(i)\log(\upsilon(1,th)) - \sum_{L}^{i=th} ih(i)\log(\upsilon(th,L+1)). \tag{3}$$

Through minimizing the cross-entropy, the optimal thresholds are achieved, denoted as

$$th^* = \operatorname{argmin}\left(f_{cross}(th)\right). \tag{4}$$

In Eq. (3), the multilevel method is for the vector $th = [th_1, th_2, \ldots, th_n]$, which can be calculated as

$$f_{cross}(th) = \sum_{L}^{i=1} ih(i)\log(i) - \sum_{n}^{i=1} H_i, \tag{5}$$

where $n$ denotes the threshold number and $H_i$ is calculated as:

$$H_1 = \sum_{th_1-1}^{i=1} ih(i)\log(\upsilon(1,th_1)), \tag{6}$$

$$H_k = \sum_{th_k-1}^{i=th_k-1} ih(i)\log(\upsilon(th_{k-1},th_k)), 1 < k < n, \tag{7}$$

$$H_n = \sum_{L}^{i=th_n} ih(i)\log(\upsilon(th_n,L+1)). \tag{8}$$

## GJO algorithm

The GJO algorithm is a novel approach inspired by the behaviors of golden jackals (*Chopra & Mohsin Ansari, 2022*). Similar to other MA algorithms, the GJO provides a method for addressing practical optimization problems. Golden jackals engage in collaborative hunting, typically in pairs (males and females) or groups. For golden jackals, the hunting process can be described as the exploration and exploitation phase (*Yang & Wang, 2024*). The mathematical models for these processes are described below.

### Exploration phase

In the exploration phase, the golden jackal utilizes its unique ability to locate and stalk the prey. Male and female jackals hunt for prey together. The mathematical model of behavior is described as:

$$P_1(t) = P_M(t) - E \cdot |P_M(t) - rl \cdot Prey(t)|, \tag{9}$$

$$P_2(t) = P_{FM}(t) - E \cdot |P_{FM}(t) - rl \cdot Prey(t)|, \tag{10}$$

$$E = 1.5 \times E_0 \times (1 - t/T), \tag{11}$$

$$rl = 0.05 \times LF, \tag{12}$$

where $P_M(t)$ and $P_{FM}(t)$ represent the positions of the male and female jackals, respectively. $Prey(t)$ represents the position vector of the prey, $E$ denotes the prey's escape energy, $E_0$

represents the initial escape energy in [-1,1], $t$ is the current iteration, $T$ is the maximum iteration, $rl$ is a random vector, and $LF$ denotes the Lévy flight function (*Heidari et al., 2019*).

Finally, the positions of the golden jackal are updated with the positions of the male and female jackals, represented as:

$$P(t+1) = 0.5 \times (P_1(t) + P_2(t)). \tag{13}$$

### *Exploitation phase*

When the prey is tracked and pursued by golden jackals, the prey's evasion energy degrades rapidly, and a pair of jackals surround the prey. After encircling the prey, the jackals pounce and capture it. The behavior of jackals can be described as follows.

$$P_1(t) = P_M(t) - E \cdot |rl \cdot P_M(t) - Prey(t)|, \tag{14}$$

$$P_2(t) = P_{FM}(t) - E \cdot |rl \cdot P_{FM}(t) - Prey(t)|. \tag{15}$$

Finally, the positions of the golden jackal are also updated using the mean position with Eq. (13).

## DE algorithm

The DE algorithm is a simple and effective population-based method, which is realized by mutating the differences between randomly selected pairs of target vectors (*Storn & Price, 1997*). Through these individual differences, the DE algorithm is guided to search for the optimal value. It primarily consists of mutation, crossover, and selection operations, and the processes can be expressed as follows (*Wunnava et al., 2022*).

### *Mutation operation*

In the population, mutation vectors are generated through the mutation operation. The commonly used mutation operator is denoted as:

$$V_i(t) = X_{best}(t) + FM \cdot (X_{r1}(t) - X_{r2}(t)), \tag{16}$$

where $FM$ is the scaling control parameter in [0, 2], $X_{best}(t)$ represents the individual vector for t iterations, $r_1$ and $r_2$ are randomly selected values from the population.

### *Crossover operation*

After generating the mutant vector, a crossover operation is performed on the source vector and its corresponding mutation vector to generate the crossover vector. This process is mathematically represented as:

$$U_i(t) = \begin{cases} V_i(t), if\,(rand \le CR) \\ X_i(t), otherwise \end{cases}, \tag{17}$$

where $CR$ denotes a crossover factor in [0, 1].

*Selection operation*

If the fitness value of the crossover vector is better than that of the source individual vector, the individual vector is updated to the crossover vector. Otherwise, the individual vector remains unchanged. The selection operation is denoted as:

$$X_i(t+1) = \begin{cases} U_i(t), f(U_i(t)) < f(X_i(t)) \\ X_i(t), otherwise \end{cases}, \tag{18}$$

where $f(\cdot)$ represents the fitness function.

## The proposed DEGJO algorithm

As mentioned above, many studies have demonstrated the GJO algorithm's remarkable search capabilities with a simple structure (*Lou et al., 2024*). In the GJO algorithm, golden jackals exhibit hunting behavior in pairs, typically led by male jackals with females following suit. However, this strategy may cause the algorithm to converge to a local optimum. On the other hand, the DE algorithm quickly searches for the minimum region in the search space and is easily integrated with other MA algorithms. In this article, we propose a hybrid DEGJO algorithm based on the DE and the GJO algorithms. This integration aims to further optimize the position of the jackal, enhancing its ability to escape local optimal solutions.

The GJO algorithm is combined with DE with mutation, crossover, and selection mechanisms. After the optimal position obtained by the GJO algorithm, the mutation, crossover, and selection operations of the DE are used to determine the best position of the golden jackal, and the corresponding fitness value is calculated. If the fitness value after differential evolution is better than that of the optimal position of the jackal, the differential evolution position is used as the optimal position. Otherwise, the optimal position of the golden jackal remains unchanged. The pseudo-code of the DEGJO algorithm is displayed in Algorithm 1.

## Computational complexity of the DEGJO

Computational complexity is closely related to the dimension number and running time, and the level of computational complexity directly affects the efficiency of an algorithm. The big-O notation provides a reliable method for quantifying and assessing the stability of algorithms (*Pan et al., 2023*).

The main parameters of the algorithm include the population size $(N)$, maximum iteration number $(T)$, and dimension number $(D)$. According to the optimization process of the GJO algorithm, the computational complexity is calculated as follows (*Chopra & Mohsin Ansari, 2022*):

$$O(\text{GJO}) = O(N \times (1 + T + T \times D)). \tag{19}$$

For the DEGJO algorithm, the computational complexity is based on the GJO algorithm with the addition of the complexity of the DE algorithm. The computational complexity of the DE algorithm is $O(T \times N)$. Therefore, the computational complexity is represented as

$$O(\text{DEGJO}) = O(N \times (1 + 2 \times T + T \times D)). \tag{20}$$

---

**Algorithm 1** Pseudo-code of DEGJO algorithm

---

**Input:** The population size $N$ and maximum iteration number $T$
**Output:** The optimal solution
 1: Initialize the random prey
 2: **while** $t < T$ **do**
 3:    Calculate the fitness values of preys
 4:    Set $P_1$ as the first prey (Male Jackal position)
 5:    Set $P_2$ as the second prey (Female Jackal position)
 6:    **for**  each prey **do**
 7:       Update the escape energy $E$ with Eq.(11)
 8:       Update $rl$ with Eq.(12)
 9:       **if** $E >= 1$ **then**
10:          Update the Exploration phase position by Eqs.(9) and (10)
11:       **else**
12:          Update the Exploitation Phase Position by Eqs.(14) and (15)
13:       **end if**
14:       The jackal positions are updated with Eq.(13)
15:       The obtained position of the golden jackal is updated by differential evolution with Eqs. (16)-(18)
16:       Compare the position after differential evolution with the optimal position of the golden jackal and select the position with with better value of the fitness function as the optimal position of the golden jackal.
17:    **end for**
18:    $t = t + 1$
19: **end while**
20: Return $P_1$

---

**Scheme 1    Algorithm 1: Pseudo-code of the DEGJO algorithm.**

## DEGJO-based multilevel thresholding method

The DEGJO algorithm is used to search for multilevel thresholding through cross-entropy minimization. The flowchart in Fig. 1 displays the DEGJO algorithm for multilevel thresholding. The detailed steps are described below:

Step 1: Input a grayscale image $I_{gray}$ and calculate the corresponding histogram of the image $h_{gray}$.

Step 2: The main parameters are set: $N$, $T$, the number of thresholds $k$, and the search range within $[0, 255]$.

Step 3: The initial positions of the prey are randomly generated and the corresponding fitness values are calculated using Eq. (4). Then, comparing the fitness values, the positions of the best prey and second-best prey are set as $Y_1$ and $Y_2$, respectively.

Step 4: Calculate the prey energy $E$ using Eq. (11). Depending on the value of $E$, the positions of the prey are updated through the exploration phase (Eqs. (9) and (10)) and exploitation phase (Eqs. (14) and (15)).

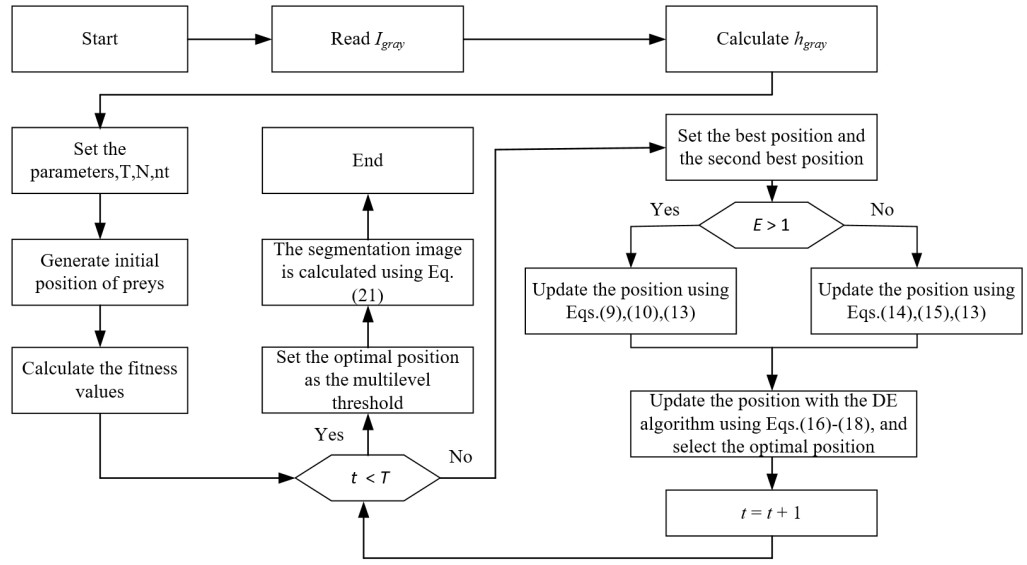

**Figure 1   The flowchart of the DEGJO algorithm for multilevel threshold image segmentation.**

Step 5: The DE algorithm is used to update the obtained optimal position of the jackal with Eqs. (16)–(18), and the corresponding fitness value is calculated. Determine whether to update the optimal position of the jackal through the fitness value.

Step 6: If the iteration number reaches $T$, the optimal position of the golden jackal is used as the optimal thresholding.

Step 7: The segmentation image $I_s$ is achieved with the optimal thresholding, represented as:

$$I_s(x,y) = \begin{cases} 0, & if \ I_{gray}(x,y) < th_1 \\ th_{i-1}, & if \ th_{i-1} < I_{gray}(x,y) < th_i, i = 2,3,\ldots,k-1 \\ th_k, & if \ I_{gray}(x,y) > th_k \end{cases} \tag{21}$$

# EXPERIMENTAL ANALYSIS AND DISCUSSION

In this section, experiments are performed to prove the algorithm's performance using the CEC2021 benchmark function and test images. Other MA algorithms such as DE (*Storn & Price, 1997*), GWO (*Mirjalili, Mirjalili & Lewis, 2014*), WOA (*Mirjalili & Lewis, 2016*), GJO (*Chopra & Mohsin Ansari, 2022*) and HGJO (*Lou et al., 2024*) are executed experimentally to compare their performance with the proposed algorithm. The parameters of the relevant algorithms are depicted in Table 1.

## Benchmark function testing
### Test functions and experiment settings
Multiple experiments are performed using the CEC2021 benchmark function (*Lou et al., 2024*; *Mohamed et al., 2022*; *Wang et al., 2023b*) to verify the DEGJO algorithm's performance. Table 2 displays the detailed information on the benchmark functions.

**Table 1  Parameters of different meta-heuristic algorithms.**

| Algorithms | Parameters value |
|---|---|
| DE | $FM=0.5, CR=0.9$ |
| GWO | $E_1 = [2, 0]$ |
| WOA | $E_1 = [2, 0]$ |
| GJO | $E_1 = [1.5, 0]$ |
| HGJO | $\theta = [0°, 360°]$ |
| | $E_1 = [1.5, 0]$ |
| DEGJO | $FM=0.5, CR=0.9$ |
| | $E_1 = [1.5, 0]$ |

For the previously mentioned algorithm, the population number, dimension number, and maximum iteration number are set to 30, 10, and 200, respectively.

### Analysis of the results of the CEC2021 benchmark functions

To analyze the results, two measures of average fitness (Avg) and standard deviation (Std) are utilized. After 20 independent runs, the measurement results of different MA algorithms are depicted in Table 3. Figure 2 illustrates the convergence curves of the DEGJO and other MA algorithms.

Table 3 illustrates that the DEGJO algorithm obtains the best results in the test functions compared to the other algorithms. Specifically, for F2, F3, F4, F6, and F8, both the DEGJO and HGJO algorithms achieve the minimum theoretical values, outperforming the results of the other algorithms. Although the results of the DEGJO algorithm are not optimal in F1, F7, F9, and F10, there are still significant improvements with the GJO and HGJO algorithms. These results highlight the superior search capabilities of the proposed algorithm.

As observed in Fig. 2, the DEGJO algorithm exhibits excellent convergence ability. In the unimodal function F1, the convergence accuracy of the DEGJO algorithm is better than other algorithms and the HGJO algorithm. For the basic function F2–F4, the DEGJO algorithm shows good searching ability and has almost the same convergence performance as the HGJO algorithm. For the functions F5–F7 and F8–F10, the DEGJO algorithm consistently outperforms the other MA algorithms in terms of search results.

### Segmented image testing

#### Test images

To assess the adaptability, the DEGJO is employed for image segmentation. Experiments are performed on six test images used in the literature (*Houssein et al., 2021b*; *Wang & Song, 2022*). These images are selected from the Open Standard Test Dataset and Berkeley Segmentation Dataset (*Martin et al., 2001*), as depicted in Fig. 3.

Figure 3 shows the benchmark images used for the testing along with their corresponding histograms. These histograms reveal variations of gray-level intensity in different images, which facilitate the verification of the robustness and applicability of the proposed method on different datasets.

**Table 2   CEC2021 benchmark functions (*Lou et al., 2024*).**

|  | Functions number | $f_{min}$ | Search range |
|---|---|---|---|
| Unimodal Function | F1 | 0 |  |
| Basic Functions | F2 | 0 |  |
|  | F3 | 0 |  |
|  | F4 | 0 |  |
| Hybrid Functions | F5 ($N = 3$) | 0 | $[-100, 100]$ |
|  | F6 ($N = 4$) | 0 |  |
|  | F7 ($N = 5$) | 0 |  |
| Composition Functions | F8 ($N = 3$) | 0 |  |
|  | F9 ($N = 4$) | 0 |  |
|  | F10 ($N = 5$) | 0 |  |

**Table 3   Results of the different algorithms.**

| Function |  | DE | GWO | WOA | GJO | HGJO | DEGJO |
|---|---|---|---|---|---|---|---|
| F1 | Avg | 3.75E+00 | 2.23E−15 | 8.05E−23 | 7.48E−35 | 3.27E−183 | **2.61E−235** |
|  | Std | 2.06E+00 | 2.53E−15 | 2.10E−22 | 2.68E−34 | 0.00E+00 | **0.00E+00** |
| F2 | Avg | 7.50E+01 | 4.36E+01 | 4.29E+02 | 3.92E−03 | **0.00E+00** | **0.00E+00** |
|  | Std | 3.65E+01 | 8.26E+01 | 5.05E+02 | 1.84E−02 | **0.00E+00** | **0.00E+00** |
| F3 | Avg | 2.04E+01 | 3.74E+01 | 3.61E+00 | 1.26E+00 | **0.00E+00** | **0.00E+00** |
|  | Std | 2.19E+00 | 2.61E+01 | 9.62E+00 | 5.93E+00 | **0.00E+00** | **0.00E+00** |
| F4 | Avg | 1.75E+00 | 1.01E+00 | 8.12E−03 | 8.13E−02 | **0.00E+00** | **0.00E+00** |
|  | Std | 3.27E−01 | 8.40E−01 | 2.51E−02 | 3.55E−01 | **0.00E+00** | **0.00E+00** |
| F5 | Avg | 2.35E+00 | 3.67E+00 | 2.88E−07 | 5.31E−04 | 6.11E−222 | **0.00E+00** |
|  | Std | 1.09E+00 | 6.52E+00 | 1.29E−06 | 2.38E−03 | 0.00E+00 | **0.00E+00** |
| F6 | Avg | 1.74E+00 | 2.64E+00 | 2.32E+01 | 1.66E−01 | **0.00E+00** | **0.00E+00** |
|  | Std | 3.59E−01 | 2.42E+00 | 5.95E+01 | 5.05E−01 | **0.00E+00** | **0.00E+00** |
| F7 | Avg | 9.39E−01 | 7.04E+00 | 1.36E−01 | 6.12E−02 | 3.04E−07 | **1.53E−87** |
|  | Std | 2.49E−01 | 2.92E+01 | 3.04E−01 | 2.28E−01 | 8.95E−07 | **6.52E−87** |
| F8 | Avg | 2.86E−01 | 6.67E−13 | 7.79E+01 | **0.00E+00** | **0.00E+00** | **0.00E+00** |
|  | Std | 4.54E−01 | 2.68E−12 | 3.47E+02 | **0.00E+00** | **0.00E+00** | **0.00E+00** |
| F9 | Avg | 1.22E−02 | 1.43E−10 | 2.65E−01 | 9.77E−15 | 9.32E−129 | **1.20E−225** |
|  | Std | 3.86E−03 | 1.14E−10 | 1.18E+00 | 2.73E−15 | 4.17E−128 | **0.00E+00** |
| F10 | Avg | 5.05E+01 | 5.39E+01 | 7.96E−02 | 2.94E+01 | 4.61E−04 | **1.84E−04** |
|  | Std | 1.26E+00 | 8.56E+00 | 3.14E−02 | 3.45E+01 | 3.30E−04 | **1.47E−04** |

**Notes.**
The best values are highlighted in bold.

### *Evaluation metrics*

For image thresholding segmentation of the DEGJO algorithm, multiple evaluation metrics are required to assess the performance of the proposed algorithm. These evaluation metrics are described as follows.

   1. Average (Avg) and standard deviation (Std) values of the fitness function

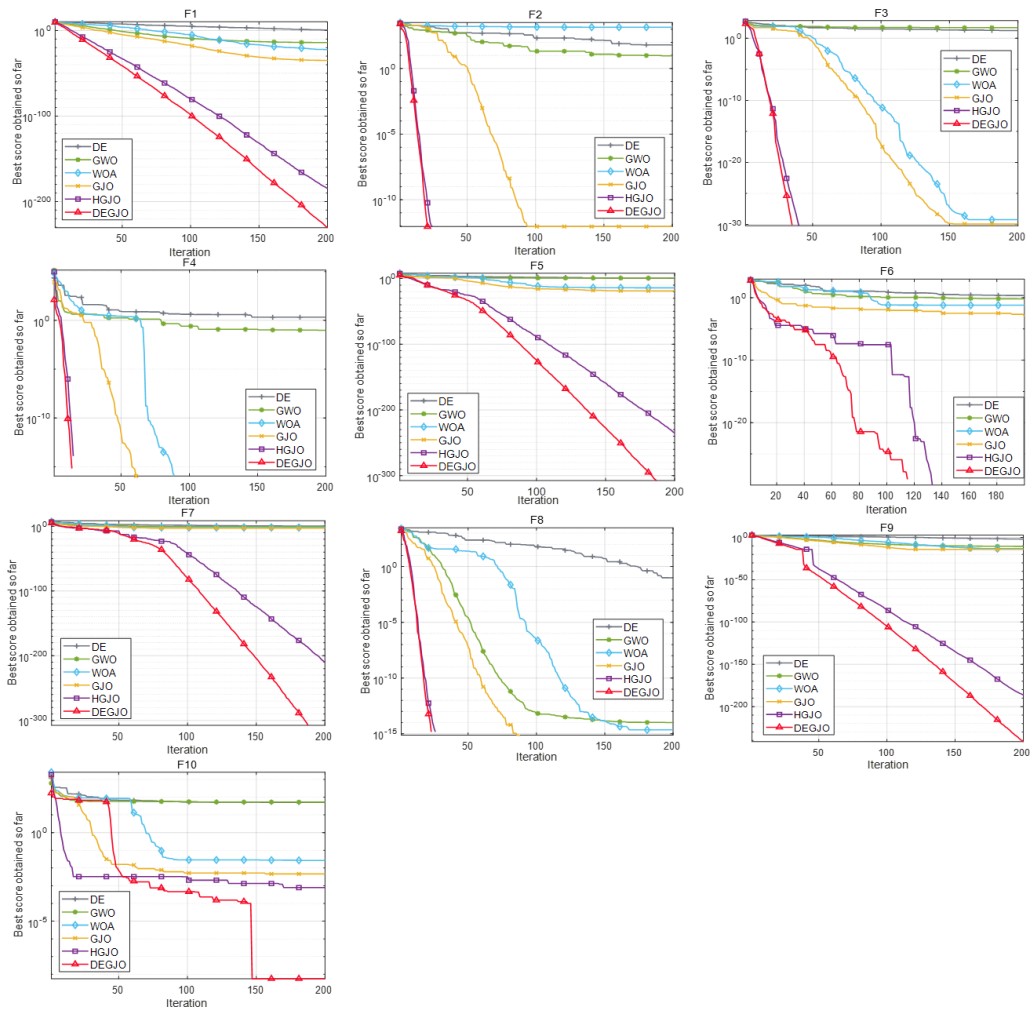

**Figure 2  Convergence curves of different meta-heuristic algorithms.**

The convergence accuracy of the algorithm is influenced by the fitness value. Accurate image segmentation thresholds can be obtained by iterating through the minimization of the fitness value. Therefore, the efficiency of the algorithm is assessed using Avg and Std of the fitness function.

2. The best threshold values

Optimization algorithms are employed for segmentation images, where the optimal thresholds are derived through iterative optimization of the algorithm. The selection of the best threshold is crucial for image segmentation, as it directly affects the quality of the segmentation.

3. Peak Signal-to-Noise Ratio (PSNR)

The PSNR denotes the peak signal-to-noise ratio of the source $I_{sou}$ and the segmented image $I_{seg}$ and is closely related to the quality of the segmented image (*Houssein et al.,*

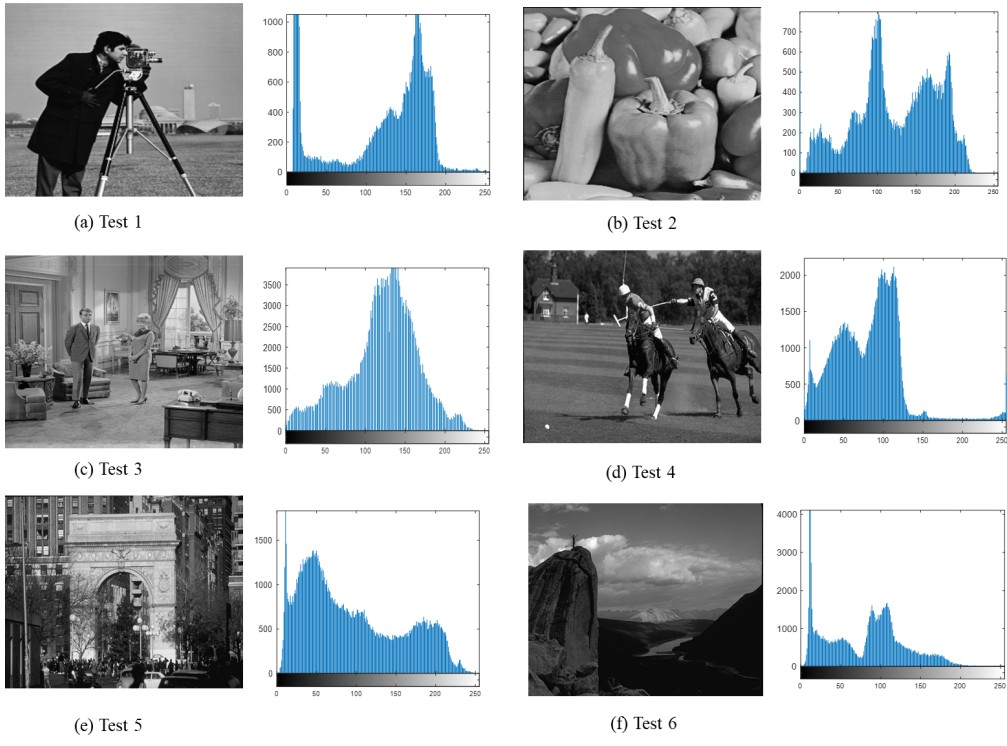

**Figure 3** (A–F) **Test images used in our experiments.** (A–C) image source credit: (A) https://github.com/mohammadimtiazz/standard-test-images-for-Image-Processing/blob/master/standard_test_images/cameraman.tif; (B) https://github.com/mohammadimtiazz/standard-test-images-for-Image-Processing/blob/master/standard_test_images/peppers_gray.tif; (C) https://github.com/mohammadimtiazz/standard-test-images-for-Image-Processing/blob/master/standard_test_images/livingroom.tif. (D–F) image source credit: (*Martin et al., 2001*).

*2022*). The PSNR can be calculated as:

$$PSNR = 10\,\log 10\left(\frac{255^2}{Mean_{SE}}\right), \tag{22}$$

where $Mean_{SE}$ is the mean square error.

4. Structural Similarity Index (SSIM)

The SSIM represents the similarity between the $I_{sou}$ and $I_{seg}$ images (*Houssein et al., 2021b*). The higher the value of SSIM, the higher the similarity between the source and the segmented image, which means that the segmentation of the image is more effective. The SSIM can be calculated as:

$$SSIM(x,y) = \frac{\left(2\mu_{sou}\mu_{seg} + c_1\right)\left(2\sigma_{ss} + c_2\right)}{\left(\mu_{sou}^2 + \mu_{seg}^2 + c_1\right)\left(\sigma_{sou}^2 + \sigma_{seg}^2 + c_2\right)}, \tag{23}$$

where $\mu_{sou}$ and $\mu_{seg}$ are the mean intensity of the $I_{sou}$ and $I_{seg}$ images, respectively. $\sigma_{sou}$ and $\sigma_{seg}$ represent the standard deviation of the $I_{sou}$ and $I_{seg}$ images, respectively. $\sigma_{ss}$ is the covariance. $c_1$ and $c_2$ are the constant numbers.

5. Feature Similarity Index (FSIM)

The FSIM denotes the similarity between the $I_{sou}$ and $I_{seg}$ images (*Aziz, Ewees & Hassanien, 2017*). A high FSIM value denotes good performance of the image segmentation method. The calculation formula is denoted as follows:

$$\text{FSIM} = \frac{\sum_{x \in \Omega} S_L(x) PC_{\max}(x)}{\sum_{x \in \Omega} PC_{\max}(x)}, \tag{24}$$

$$S_L(x) = S_{PC}(x) S_G(x), \tag{25}$$

$$S_{PC}(x) = \frac{2PC_1(x)PC_2(x) + T_1}{PC_1^2(x) + PC_2^2(x) + T_1}, \tag{26}$$

$$S_G(x) = \frac{2G_1(x)G_2(x) + T_2}{G_1^2(x) + G_2^2(x) + T_2}, \tag{27}$$

$$PC_{\max}(x) = \max\{PC_1(x), PC_2(x)\}, \tag{28}$$

where $T_1$ and $T_2$ are constant numbers. $PC(x)$ denotes the phase congruence, and $G(x)$ denotes the magnitude of the image gradient.

### Analysis of the segmented image results

The DEGJO algorithm is employed to search for the optimal multilevel threshold, with an objective function of cross-entropy minimization. For each algorithm, the $N$ is selected as 30, $T$ is chosen as 100, and the threshold levels are set to 3, 5, and 8, respectively. To reduce the randomness of the algorithms, each algorithm is run independently 20 times. The experimental results of the DEGJO and other MA algorithms are displayed in Tables 4–8.

Table 4 shows the Avg and Std of the cross-entropy values for the DEGJO and other algorithms. The best values are highlighted in bold. Compared to DE, GWO, WOA, GJO and HGJO algorithms, the DEGJO algorithm obtains the best average values in most images at all levels.

Table 5 depicts the best threshold values with the DE, GWO, WOA, GJO, HGJO, and DEGJO algorithms. Generally, most algorithms exhibit similar values when the threshold levels are 3 and 5, however, significant differences emerge in the threshold values obtained when the threshold level is 8, particularly with the HGJO algorithm.

Table 6 displays the mean PSNR values of the algorithm mentioned above. The Optimal values are marked in bold. In most images, the segmented image using the DEGJO algorithm has a higher PSNR value than other algorithms. However, for some images, other algorithms perform better than the DEGJO algorithm.

Table 7 compares the mean SSIM values derived from the performance of various algorithms. The best results are highlighted in bold, indicating superior segmentation of the original image. It can be observed that the DEGJO algorithm achieves significant results across most of the images.

Table 8 shows the mean FSIM values by employing the proposed DEGJO and other MA algorithms. The optimal values are highlighted in bold, indicating superior segmentation quality, with higher values reflecting better performance of the thresholding method.

Finally, Figs. 4–9 depict the segmentation images using the DEGJO approach at three different levels ($k = 3$, 5, and 8), alongside their respective histograms. In these figures, the

**Table 4  Results of the Avg and Std of fitness values.**

| Images | k | | DE | GWO | WOA | GJO | HGJO | DEGJO |
|--------|---|---|-----|------|------|------|------|--------|
| Test 1 | 3 | Avg | **0.7611** | 0.7616 | **0.7611** | **0.7611** | 0.7639 | **0.7611** |
| | | Std | **0** | 6.17E−04 | **0** | **0** | 3.95E−03 | **0** |
| | 5 | Avg | 0.4087 | 0.4084 | **0.4023** | 0.4060 | 0.4097 | 0.4043 |
| | | Std | 3.37E−3 | 4.13E−03 | **2.55E−03** | 4.01E−03 | 8.12E−03 | 2.02E−03 |
| | 8 | Avg | 0.2091 | 0.2082 | 0.2084 | 0.2089 | 0.2182 | **0.2078** |
| | | Std | 2.17E−3 | 8.10E−03 | 3.47E−03 | 5.97E−03 | 1.11E−02 | **1.71E−03** |
| Test 2 | 3 | Avg | 1.2190 | 1.2187 | 1.2186 | 1.2186 | 1.2255 | **1.2185** |
| | | Std | 2.87E−4 | 3.40E−4 | 1.20E−4 | 9.66E−5 | 2.12E−2 | **0** |
| | 5 | Avg | 0.5845 | 0.5836 | 0.5835 | 0.5930 | 0.5910 | **0.5811** |
| | | Std | 2.64E−03 | 3.03E−03 | 8.49E−03 | 1.98E−02 | 1.26E−02 | **5.71E−04** |
| | 8 | Avg | 0.2877 | 0.2750 | 0.2938 | 0.2903 | 0.2918 | **0.2745** |
| | | Std | 9.28E−03 | 1.05E−2 | 1.81E−02 | 1.63E−02 | 2.00E−02 | **5.34E−03** |
| Test 3 | 3 | Avg | 1.1702 | 1.1693 | 1.1694 | 1.1693 | 1.1713 | **1.1690** |
| | | Std | 1.27E−03 | 4.09E−4 | 4.53E−04 | 3.81E−04 | 4.74E−03 | **0** |
| | 5 | Avg | 0.5446 | 0.5431 | 0.5410 | 0.5461 | 0.5504 | **0.5389** |
| | | Std | 4.25E−03 | 3.67E−03 | 2.42E−03 | 6.21E−03 | 1.02E−2 | **3.03E−04** |
| | 8 | Avg | 0.2753 | 0.2624 | 0.2552 | 0.2599 | 0.2661 | **0.2536** |
| | | Std | 8.41E−03 | 9.21E−03 | 2.96E−3 | 9.40E−3 | 5.20E−3 | **1.09E−03** |
| Test 4 | 3 | Avg | 1.3467 | 1.3466 | **1.3465** | 1.3466 | 1.3468 | **1.3465** |
| | | Std | 0 | 9.49E−05 | **0** | 9.49E−05 | 2.53E−04 | **0** |
| | 5 | Avg | 0.5780 | 0.5804 | 0.5770 | 0.5769 | 0.5846 | **0.5765** |
| | | Std | 1.32E−04 | 7.44E−03 | 7.34E−04 | 5.93E−04 | 1.09E−02 | **3.16E−05** |
| | 8 | Avg | 0.2768 | 0.2762 | 0.2719 | 0.2735 | 0.2791 | **0.2709** |
| | | Std | 3.52E−03 | 5.05E−03 | 1.45E−03 | 2.95E−03 | 3.76E−03 | **0** |
| Test 5 | 3 | Avg | 1.4460 | 1.4460 | 1.4460 | 1.4460 | 1.4463 | **1.3473** |
| | | Std | 0 | 0 | 0 | 0 | 3.87E−04 | **3.39E−04** |
| | 5 | Avg | 0.6681 | 0.6675 | 0.6675 | 0.6675 | 0.6714 | **0.5833** |
| | | Std | 1.01E−03 | 0 | 0 | 0 | 3.10E−03 | 2.01E−03 |
| | 8 | Avg | 0.2983 | 0.2964 | 0.2957 | 0.2972 | 0.3086 | **0.2821** |
| | | Std | 3.29E−03 | 8.57E−04 | 0 | 1.41E−3 | 8.53E−03 | 2.34E−03 |
| Test 6 | 3 | Avg | 0.9534 | 0.9529 | **0.9525** | 0.9526 | 0.9530 | **0.9525** |
| | | Std | 8.84E−04 | 0.15E−04 | **0** | 1.55E−04 | 6.76E−04 | **0** |
| | 5 | Avg | 0.4396 | 0.4397 | 0.4392 | 0.4394 | 0.4443 | **0.4391** |
| | | Std | 3.20E−04 | 8.99E−04 | 1.05E−04 | 3.40E−04 | 8.05E−03 | **0** |
| | 8 | Avg | 0.2139 | 0.2103 | 0.2087 | 0.2095 | 0.2191 | **0.2080** |
| | | Std | 3.36E−03 | 3.36E−03 | 1.90E−03 | 2.29E−03 | 7.60E−03 | **1.58E−04** |

**Notes.**
The best values are highlighted in bold.

optimal thresholds are indicated by red vertical lines. As shown in Figs. 4–9, the contrast quality of the images improves significantly with the increasing number of thresholds.

Meng et al. (2024), *PeerJ Comput. Sci.*, DOI 10.7717/peerj-cs.2121

**Table 5  The best threshold values.**

| Images | k | DE | GWO | WOA | GJO | HGJO | DEGJO |
|---|---|---|---|---|---|---|---|
| Test 1 | 3 | 30,83,144 | 30,83,144 | 30,83,144 | 30,83,144 | 30,83,144 | 30,83,144 |
|  | 5 | 27,68,110,144,174 | 27,66,109,144,172 | 28,71,115,145,172 | 28,69,113,145,172 | 26,51,93,138,168 | 28,70,114,146,172 |
|  | 8 | 23,47,74,106,132,153,174,196 | 24,50,82,112,134,155,173,203 | 15,26,51,85,120,146,169,200 | 22,46,77,108,132,154,173,207 | 22,47,79,111,133,156,175,204 | 24,46,76,107,131,150,170,204 |
| Test 2 | 3 | 43,87,139 | 44,88,139 | 44,88,139 | 44,88,139 | 44,88,139 | 44,88,139 |
|  | 5 | 21,50,86,125,169 | 22,51,86,125,169 | 23,51,87,125,169 | 42,81,112,144,176 | 23,50,86,126,169 | 23,52,86,125,169 |
|  | 8 | 2,26,49,78,98,123,154,184 | 9,29,52,79,100,125,155,182 | 10,29,53,80,101,124,153,180 | 10,27,51,79,101,125,154,181 | 9,31,55,82,100,124,152,181 | 10,30,53,79,101,124,152,180 |
| Test 3 | 3 | 43,95,145 | 44,95,145 | 44,95,145 | 44,95,145 | 43,94,145 | 44,95,145 |
|  | 5 | 32,68,103,134,168 | 34,70,103,133,167 | 33,69,103,133,167 | 31,67,103,133,168 | 17,70,108,133,166 | 33,70,102,134,167 |
|  | 8 | 18,35,56,76,102,126,150,181 | 22,47,69,89,111,131,152,181 | 22,43,66,89,112,131,152,181 | 21,42,63,88,111,131,152,181 | 15,39,66,90,110,131,150,181 | 21,42,66,90,111,132,153,182 |
| Test 4 | 3 | 34,77,152 | 35,77,152 | 35,77,152 | 35,77,152 | 34,77,151 | 35,77,152 |
|  | 5 | 20,45,73,102,165 | 21,46,73,102,165 | 21,46,73,102,165 | 21,46,73,102,165 | 21,46,73,102,165 | 21,46,73,102,165 |
|  | 8 | 16,33,49,67,87,106,135,196 | 16,33,49,67,86,106,135,194 | 16,33,49,67,87,106,135,196 | 16,32,48,66,86,106,136 | 17,36,52,69,88,107,139,205 | 16,33,49,67,87,106,135,196 |
| Test 5 | 3 | 36,78,142 | 36,78,142 | 36,78,142 | 36,78,142 | 36,78,142 | 35,77,152 |
|  | 5 | 25,49,79,119,168 | 25,49,79,119,168 | 25,49,79,119,168 | 25,49,79,119,168 | 25,48,79,119,166 | 22,46,75,103,161 |
|  | 8 | 20,35,51,70,94,123,157,191 | 19,34,50,70,95,123,159,190 | 21,36,52,71,95,123,156,190 | 21,36,52,71,95,123,156,190 | 20,35,51,71,96,123,155,187 | 18,33,48,66,86,107,138,185 |
| Test 6 | 3 | 28,72,128 | 28,72,128 | 28,72,128 | 28,72,128 | 28,72,128 | 28,72,128 |
|  | 5 | 21,42,72,105,141 | 21,42,73,105,141 | 21,42,73,105,141 | 21,42,73,105,141 | 21,42,73,105,141 | 21,42,73,105,141 |
|  | 8 | 18,33,51,75,99,117,138,165 | 18,32,50,73,98,117,140,169 | 18,32,50,73,98,117,140,169 | 18,32,50,74,98,117,141,170 | 19,33,51,74,98,116,135,167 | 18,32,50,74,98,117,140,168 |

**Table 6  Results of the mean PSNR.**

| Images | k | DE | GWO | WOA | GJO | HGJO | DEGJO |
|--------|---|------|------|------|------|------|-------|
| Test 1 | 3 | **18.7967** | 18.7582 | 18.7127 | 18.7469 | 18.7611 | 18.7184 |
|        | 5 | 22.4806 | 22.4347 | 22.8284 | 22.5989 | 22.4820 | **22.8473** |
|        | 8 | 25.5671 | **25.6409** | 25.2934 | 25.5776 | 25.5854 | 25.6030 |
| Test 2 | 3 | 17.8345 | 17.8340 | 17.8277 | 17.8317 | 17.8102 | **17.8399** |
|        | 5 | 21.2451 | 21.3074 | 21.2425 | **21.4339** | 21.2702 | 21.2628 |
|        | 8 | 24.7374 | 24.9890 | 24.7006 | **25.0026** | 24.6506 | 23.8157 |
| Test 3 | 3 | **18.5074** | 18.4300 | 18.4107 | 18.4048 | 18.4689 | 18.4347 |
|        | 5 | 22.0869 | 22.1710 | **22.2243** | 22.0245 | 22.0935 | 22.1923 |
|        | 8 | 25.1981 | 25.3756 | **25.5581** | 25.4552 | 25.4987 | 25.5163 |
| Test 4 | 3 | 18.9777 | 18.9974 | 19.0048 | 18.9961 | 18.9750 | **19.0048** |
|        | 5 | 22.6020 | 22.6179 | 22.6522 | **22.6564** | 22.6307 | 22.6335 |
|        | 8 | 26.0114 | 26.0202 | 26.1691 | 26.0932 | 26.0439 | **26.1921** |
| Test 5 | 3 | 17.7790 | 17.7790 | 17.7790 | 17.7790 | 17.7494 | **17.9873** |
|        | 5 | **21.3595** | 21.3347 | 21.3347 | 21.3410 | 21.3160 | 20.6440 |
|        | 8 | 24.8356 | 24.8856 | **24.9075** | 24.8747 | 24.8193 | 23.6847 |
| Test 6 | 3 | 19.6786 | 19.6403 | 19.6564 | 19.6190 | **19.6818** | 19.6729 |
|        | 5 | 22.9878 | 23.0368 | 23.0054 | 22.9959 | 23.0613 | **23.0517** |
|        | 8 | 25.8374 | 26.0622 | 26.0283 | 26.0307 | 25.8926 | **26.1068** |

**Notes.**
The best values are highlighted in bold.

**Table 7  Results of the mean SSIM.**

| Images | k | DE | GWO | WOA | GJO | HGJO | DEGJO |
|--------|---|------|------|------|------|------|-------|
| Test 1 | 3 | 0.6667 | 0.6726 | 0.6724 | **0.6727** | 0.6724 | 0.6718 |
|        | 5 | 0.7137 | 0.7129 | 0.7200 | 0.7159 | 0.7111 | **0.7202** |
|        | 8 | 0.7698 | 0.7678 | **0.7818** | 0.7785 | 0.7733 | 0.7614 |
| Test 2 | 3 | **0.6912** | 0.6905 | 0.6908 | 0.6906 | 0.6889 | 0.6902 |
|        | 5 | 0.7702 | 0.7716 | 0.7716 | 0.7712 | 0.7694 | **0.7719** |
|        | 8 | 0.8356 | 0.8389 | 0.8354 | **0.8410** | 0.8338 | 0.8280 |
| Test 3 | 3 | **0.6604** | 0.6603 | 0.6603 | **0.6604** | 0.6602 | 0.6603 |
|        | 5 | 0.7581 | 0.7614 | 0.7610 | 0.7573 | 0.7598 | **0.7629** |
|        | 8 | 0.8422 | 0.8454 | **0.8496** | 0.8475 | 0.8473 | 0.8495 |
| Test 4 | 3 | 0.6244 | 0.6240 | 0.6237 | 0.6239 | **0.6249** | 0.6237 |
|        | 5 | 0.7499 | 0.7513 | 0.7514 | **0.7516** | 0.7490 | 0.7506 |
|        | 8 | 0.8464 | 0.8456 | 0.8479 | 0.8464 | 0.8465 | **0.8485** |
| Test 5 | 3 | 0.6498 | 0.6498 | 0.6498 | 0.6498 | 0.6492 | **0.6538** |
|        | 5 | 0.7879 | **0.7885** | **0.7885** | 0.7884 | 0.7861 | 0.7822 |
|        | 8 | 0.8703 | **0.8728** | 0.8716 | 0.8722 | 0.8710 | 0.8618 |
| Test 6 | 3 | 0.5381 | 0.5383 | 0.5384 | 0.5382 | 0.5383 | **0.5386** |
|        | 5 | 0.6115 | 0.6111 | 0.6107 | 0.6110 | **0.6123** | 0.6110 |
|        | 8 | **0.6722** | 0.6716 | 0.6712 | 0.6699 | 0.6683 | 0.6695 |

**Notes.**
The best values are highlighted in bold.

**Table 8 Results of the mean FSIM.**

| Images | k | DE | GWO | WOA | GJO | HGJO | DEGJO |
|--------|---|------|------|------|------|------|-------|
| Test 1 | 3 | 0.8182 | 0.8213 | 0.8209 | **0.8215** | 0.8212 | 0.8206 |
|        | 5 | 0.8752 | 0.8744 | 0.8816 | 0.8776 | 0.8748 | **0.8822** |
|        | 8 | 0.9255 | 0.9269 | 0.9186 | 0.9235 | 0.9239 | **0.9283** |
| Test 2 | 3 | **0.7477** | 0.7475 | 0.7473 | 0.7472 | 0.7462 | 0.7473 |
|        | 5 | 0.8054 | 0.8059 | 0.8051 | **0.8106** | 0.8053 | 0.8059 |
|        | 8 | 0.8678 | 0.8718 | 0.8670 | **0.8725** | 0.8669 | 0.8577 |
| Test 3 | 3 | 0.8247 | 0.8249 | 0.8245 | 0.8247 | 0.8245 | **0.8252** |
|        | 5 | 0.9092 | 0.9093 | 0.9100 | 0.9085 | 0.9079 | **0.9104** |
|        | 8 | 0.9497 | 0.9525 | 0.9537 | 0.9535 | 0.9517 | **0.9542** |
| Test 4 | 3 | 0.7305 | 0.7308 | 0.7307 | 0.7306 | **0.7319** | 0.7307 |
|        | 5 | 0.8358 | 0.8373 | 0.8371 | **0.8374** | 0.8359 | 0.8368 |
|        | 8 | 0.9096 | 0.9092 | 0.9119 | 0.9108 | 0.9094 | **0.9123** |
| Test 5 | 3 | **0.8008** | **0.8008** | **0.8008** | **0.8008** | 0.8004 | 0.7955 |
|        | 5 | 0.8772 | 0.8777 | 0.8777 | 0.8778 | **0.8780** | 0.8651 |
|        | 8 | 0.9344 | **0.9351** | 0.9349 | 0.9348 | 0.9337 | 0.9184 |
| Test 6 | 3 | 0.7985 | 0.7985 | 0.7985 | **0.7986** | 0.7985 | 0.7983 |
|        | 5 | 0.8278 | 0.8282 | 0.8281 | 0.8279 | **0.8289** | 0.8282 |
|        | 8 | 0.8746 | 0.8779 | 0.8771 | 0.8676 | 0.8740 | **0.8783** |

**Notes.**
The best values are highlighted in bold.

# CONCLUSIONS

Image segmentation is a critical step in the accurate processing and analysis of images. Various techniques using multilevel thresholding have been developed to solve this challenge. Thresholding-based segmentation methods are widely utilized for their simple operation and efficient point characteristics. In this study, we present a hybrid DEGJO algorithm, which uses the MCE as a fitness function to determine optimal threshold values. The DEGJO algorithm aims to obtain the optimal threshold using the MCE method. Experiments are conducted on benchmark functions and images, comparing the performance of the DEGJO algorithm with other metaheuristic algorithms, including DE, GWO, WOA, GJO, and HGJO. The results of these experiments highlight the superior performance of the DEGJO algorithm, evident in its outperformance across various metrics including fitness values, PSNR, SSIM and FSIM, compared to other optimization algorithms.

In future work, we will further optimize the performance of the DEGJO algorithm using actual captured images employ Otsu, Kapur entropy, Fuzzy entropy, and Masi entropy for multilevel thresholding image segmentation.

# ACKNOWLEDGEMENTS

The authors would like to thank the owners of the dataset for sharing their data.

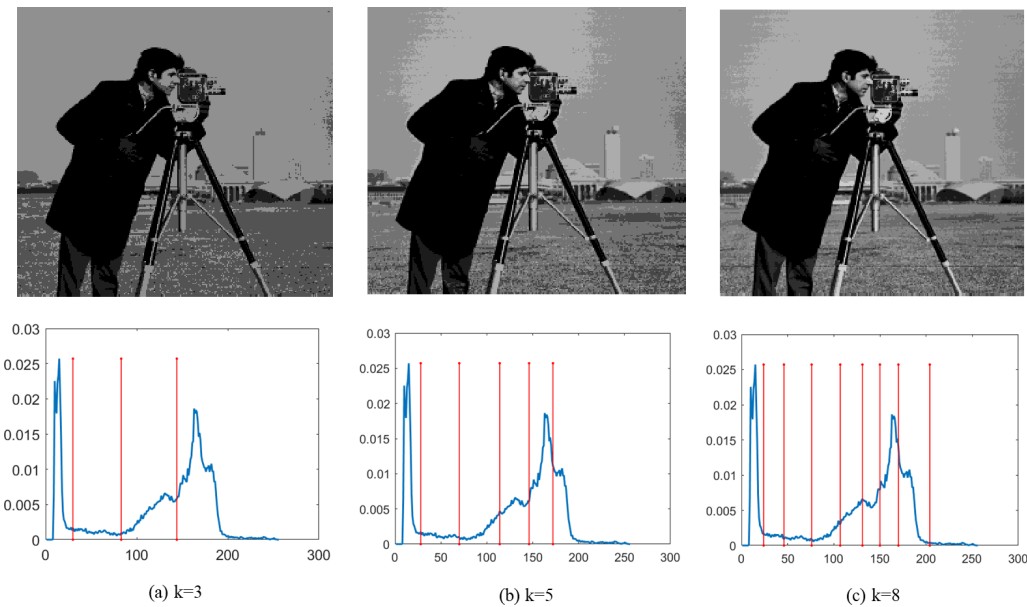

(a) k=3        (b) k=5        (c) k=8

**Figure 4** **(A–C) Segmentation results of the DEGJO to the Test 1 image.** Image source credit: https://github.com/mohammadimtiazz/standard-test-images-for-Image-Processing/blob/master/standard_test_images/cameraman.tif.

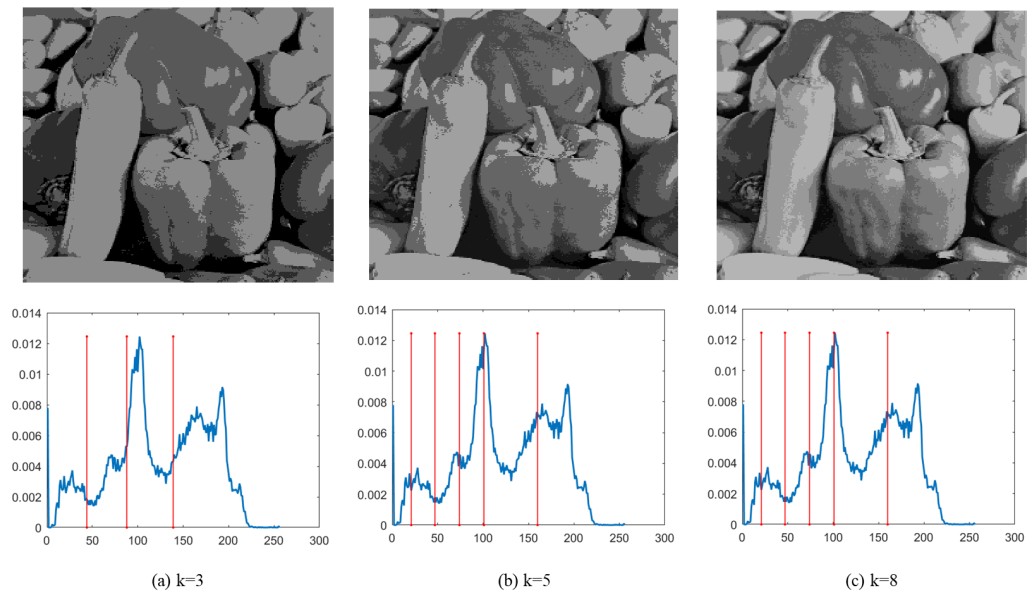

(a) k=3        (b) k=5        (c) k=8

**Figure 5** **(A–C) Segmentation results of the DEGJO to the Test 2 image.** Image source credit: https://github.com/mohammadimtiazz/standard-test-images-for-Image-Processing/blob/master/standard_test_images/peppers_gray.tif.

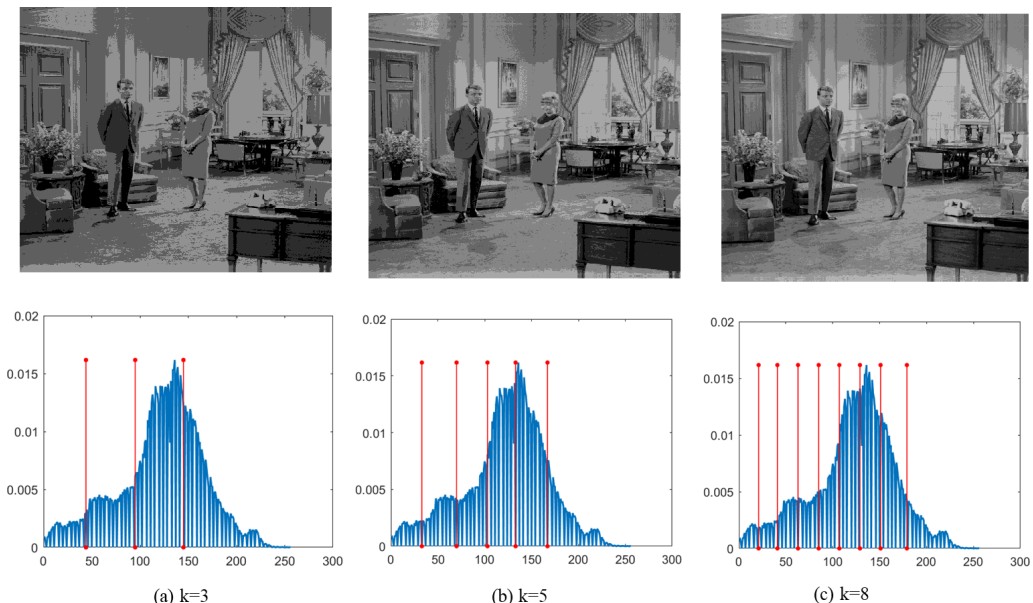

**Figure 6** **Segmentation results of the DEGJO to the Test 3 image.** Image source credit: https://github.com/mohammadimtiazz/standard-test-images-for-Image-Processing/blob/master/standard_test_images/livingroom.tif.

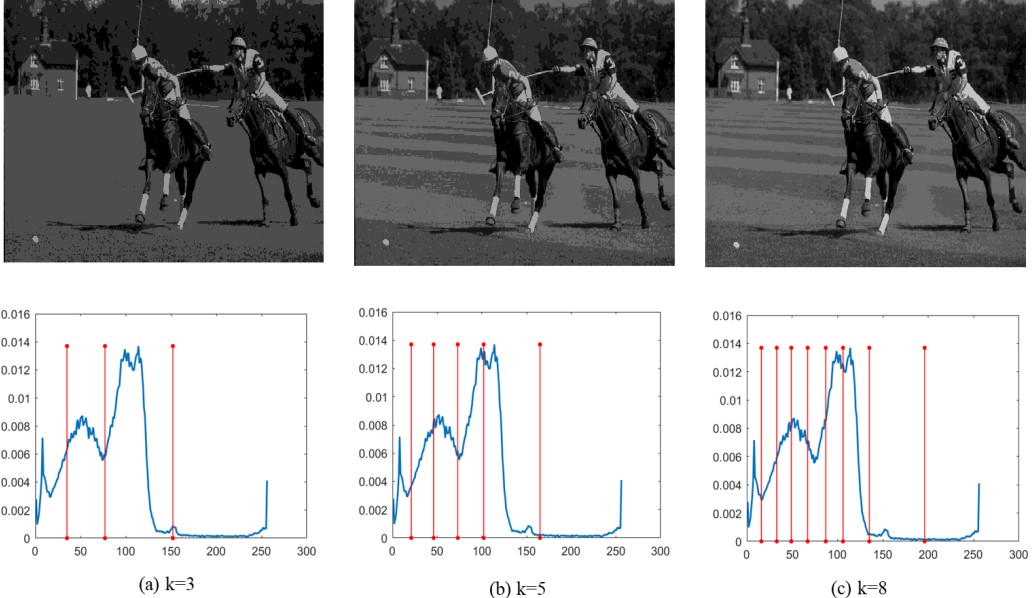

**Figure 7** **(A–C) Segmentation results of the DEGJO to the Test 4 image.** Image source credit: (*Martin et al., 2001*).

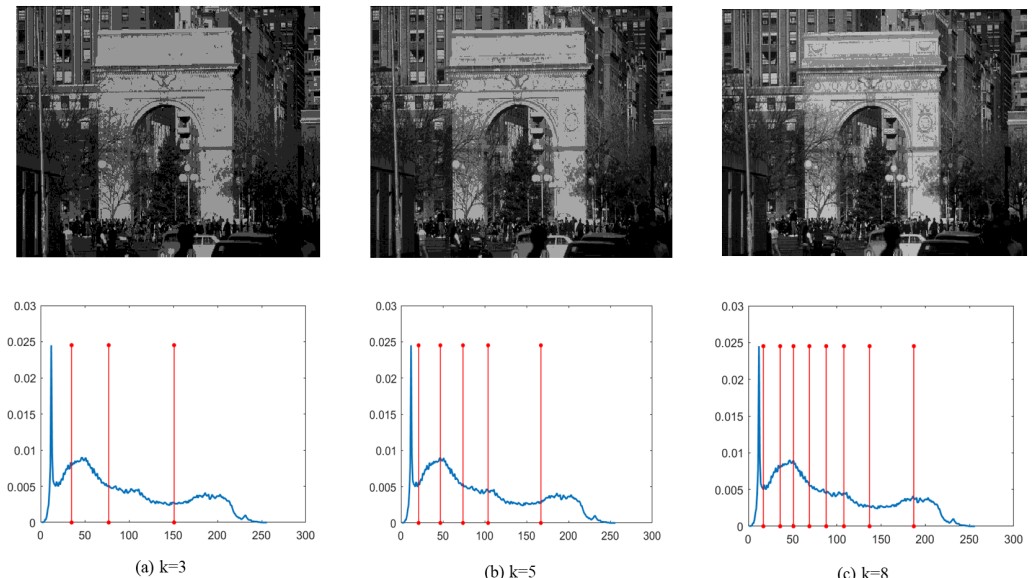

**Figure 8** **(A–C) Segmentation results of the DEGJO to the Test 5 image.** Image source credit: (*Martin et al., 2001*).

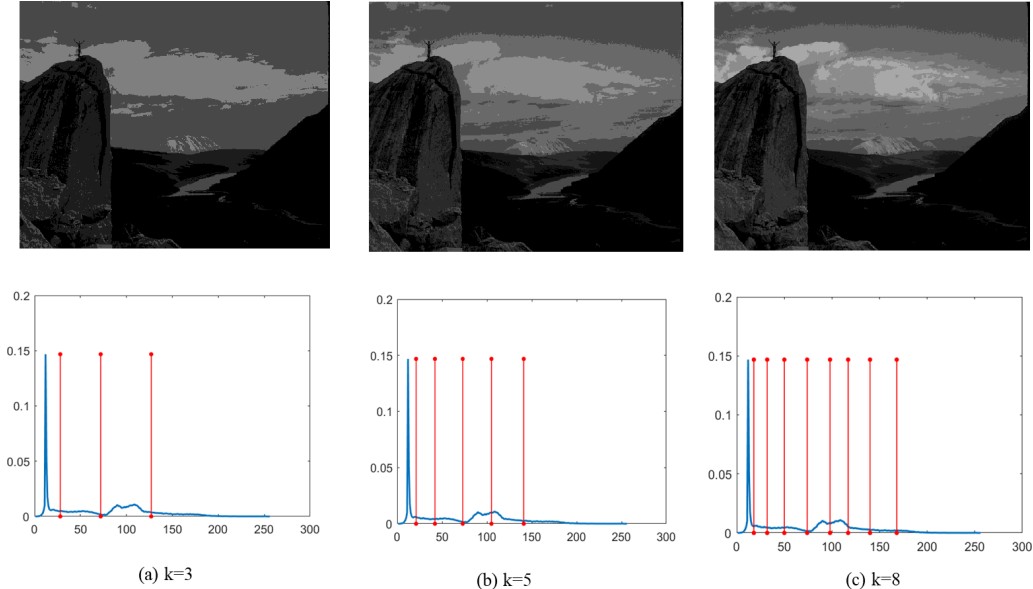

**Figure 9** **(A–C) Segmentation results of the DEGJO to the Test 6 image.** Image source credit: (*Martin et al., 2001*).

### Funding
This work was supported by the Natural Science Research Project in Anhui Province University of China (Nos. KJ2021A1164, KJ2021A1165, and 2022AH051669), and the Academic Funding Program for Top Talents in Higher Education Disciplines (No. gxbjZD2022086). The funders had no role in study design, data collection and analysis, decision to publish, or preparation of the manuscript.

### Grant Disclosures
The following grant information was disclosed by the authors:
Natural Science Research Project in Anhui Province University of China: KJ2021A1164, KJ2021A1165, 2022AH051669.
Academic Funding Program for Top Talents in Higher Education Disciplines: gxbjZD2022086.

### Competing Interests
The authors declare there are no competing interests.

### Author Contributions

- Xianmeng Meng conceived and designed the experiments, performed the experiments, analyzed the data, performed the computation work, prepared figures and/or tables, authored or reviewed drafts of the article, and approved the final draft.
- Linglong Tan conceived and designed the experiments, performed the experiments, performed the computation work, authored or reviewed drafts of the article, and approved the final draft.
- Yueqin Wang analyzed the data, prepared figures and/or tables, and approved the final draft.

### Data Availability
The code is available at GitHub and Zenodo:

– https://github.com/Mengxianm/DEGJO.

– mengxianm. (2024). mengxianm/DEGJO: v1.0.0 (DEGJO). Zenodo. https://doi.org/10.5281/zenodo.11156899.

The Open Standard Test Dataset is available at: https://github.com/mohammadimtiazz/standard-test-images-for-Image-Processing/tree/master/standard_test_images.

The Berkeley Segmentation Dataset was generated from the University of California, Berkeley and is available at: https://www2.eecs.berkeley.edu/Research/Projects/CS/vision/bsds/BSDS300/html/dataset/images.html.

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
