# Peer review of "An efficient hybrid differential evolution-golden jackal optimization algorithm for multilevel thresholding image segmentation"

_PeerJ Computer Science, doi:10.7717/peerj-cs.2121_

## Round 0.1 · original submission · Major Revisions

Dear authors,

Thank you for submitting your article. Feedback from the reviewers is now available. It is not recommended that your article be published in its current format. However, we strongly recommend that you address the issues raised by the reviewers, especially those related to readability, experimental design and validity, and resubmit your paper after making the necessary changes.

Best wishes,

**Language Note:** PeerJ staff have identified that the English language needs to be improved. When you prepare your next revision, please either (i) have a colleague who is proficient in English and familiar with the subject matter review your manuscript, or (ii) contact a professional editing service to review your manuscript. PeerJ can provide language editing services - you can contact us at [email protected] for pricing (be sure to provide your manuscript number and title). – PeerJ Staff

Reviewer 1 ·

Basic reporting

In this study, a new method for image segmentation with multiple threshold values was developed to reveal the information in the image.
It is a very exciting method, and it is considered extremely valuable to propose a hybrid method using metaheuristic methods together for the segmentation process.
I will have some suggestions to help the study reach sufficient maturity and make it easier for readers to understand.
1- It is known that there are many metaheuristic methods in the literature. In their studies, the authors preferred DE and GJO algorithms for image segmentation. By using these algorithms together, they proposed the DEGJO algorithm, which is a hybrid method. However, they did not fully mention why they chose these algorithms. What exactly is motivation?

2-DE and GJO algorithms are mentioned in the article. It is known that metaheuristic methods are created by being inspired by natural events. Optimization methods need a categorization for readers to understand. For example, Light-Based, Physics-Based, Math-Based or hybrid-based methods created by using a combination of these. These studies should be reviewed and added to the article.

3-Within the scope of this study, please state the contribution to literature and science in the introduction section without being repetitive.

Experimental design

4-No information is given about the methods and limitations of the methods used in the article. Are there any special constraints for the problem? If there is, it should be given.

5-It has been mentioned that the DEGJO algorithm avoids local solutions and converges to the global solution better than other well-known methods. How did DEGJO gain this ability? How does it strike the balance between exploration and exploitation? Please explain in detail by giving equations.

Validity of the findings

6-The conclusion part of the article should be improved and information should be given about future studies.

Additional comments

6- Correct the language, spelling, and grammatical errors used in the study.

Reviewer 2 ·

Basic reporting

The manuscript contains novel elements. However, it presents some aspects that need to be solved before reconsideration.
The authors should explicitly mention the significant contributions of the manuscript. The novelty of the paper is not highlighted.
In the introduccion section, add a paragraph with segmentation approaches that are not based on metaheuristic pronciples. There are other techniques based on artificial intelligence approaches such agent-based models “Image Segmentation by Agent-Based Pixel Homogenization”
The use of the metaheuristic algorithm (jackal optimization algorithm) is not clear. There are many metaheuristic algorithms. Explain why this algorithm was used in the context of the network training application. Is it faster? Is it easier?

Experimental design

.

Validity of the findings

.

---

## Round 0.2 · accepted · Accept

Dear authors,

Thank you for the revision and for clearly addressing all the reviewers' comments. I confirm that the paper is improved. Your paper is now acceptable for publication in light of this revision.

Best wishes,

Reviewer 1 ·

Basic reporting

All corrections have been made.

Experimental design

All corrections have been made.

Validity of the findings

All corrections have been made.

Additional comments

All corrections have been made.